# Redundant and specific roles of individual *MIR172* genes in plant development

Heng Lian[1], Long Wang[1], Ning Ma[1,2], Chuan-Miao Zhou[1], Lin Han [1,3], Tian-Qi Zhang[1], Jia-Wei Wang [1,4]*

1 National Key Laboratory of Plant Molecular Genetics, CAS Center for Excellence in Molecular Plant Sciences, Institute of Plant Physiology and Ecology, Chinese Academy of Sciences, Shanghai, China, 2 School of Life Science, Henan University, Kaifeng, China, 3 University of Chinese Academy of Sciences, Shanghai, China, 4 ShanghaiTech University, Shanghai, China

* jwwang@sippe.ac.cn

**Data Availability Statement:** All relevant data are within the paper and its Supporting Information files.

**Funding:** This work was supported by the grants from National Natural Science Foundation of China

## Abstract

Evolutionarily conserved microRNAs (miRNAs) usually have high copy numbers in the genome. The redundant and specific roles of each member of a multimember miRNA gene family are poorly understood. Previous studies have shown that the miR156-SPL-miR172 axis constitutes a signaling cascade in regulating plant developmental transitions. Here, we report the feasibility and utility of CRISPR-Cas9 technology to investigate the functions of all 5 *MIR172* family members in Arabidopsis. We show that an Arabidopsis plant devoid of miR172 is viable, although it displays pleiotropic morphological defects. *MIR172* family members exhibit distinct expression pattern and exert functional specificity in regulating meristem size, trichome initiation, stem elongation, shoot branching, and floral competence. In particular, we find that the miR156-SPL-miR172 cascade is bifurcated into specific flowering responses by matching pairs of coexpressed *SPL* and *MIR172* genes in different tissues. Our results thus highlight the spatiotemporal changes in gene expression that underlie evolutionary novelties of a miRNA gene family in nature. The expansion of *MIR172* genes in the Arabidopsis genome provides molecular substrates for the integration of diverse floral inductive cues, which ensures that plants flower at the optimal time to maximize seed yields.

## Introduction

MicroRNAs (miRNAs), a class of small single-stranded noncoding RNAs that range in length from 20 to 22 nucleotides (nt), play important roles in regulating gene expression [1–5]. It has been proposed that miRNAs originate from inverted duplication of target gene fragments, and then undergo diversification through genome-wide duplication, tandem duplication, and segmental duplication, a similar process that drives the evolution of protein gene families in plants [6–9]. As such, the ancient (i.e., evolutionarily conserved) miRNAs such as miR156, miR159/319, miR160, miR165/6, miR171, and miR172 are present in high copy numbers in the genome of *Arabidopsis thaliana* [10].

Due to the small sizes of the genes, simultaneous inactivation of all miRNA family members by generation of multiple transfer-DNA (T-DNA) mutant lines has so far been achieved for

(31788103; 31525004; 31721001; 31401026) and Strategic Priority Research Program of the Chinese Academy of Sciences (XDB27030101). The funders had no role in study design, data collection and analysis, decision to publish, or preparation of the manuscript.

**Competing interests:** The authors have declared that no competing interests exist.

**Abbreviations:** AP1, *APETALA1*; AP2, *APETALA2*; ATAC-seq, transposase-accessible chromatin sequencing; Col-0, Columbia-0; CRISPR, clustered regularly interspaced short palindromic repeats; FCA, FLOWERING CONTROL LOCUS A; FLM, FLOWERING LOCUS M; FT, *FLOWERING LOCUS T*; FUL, FRUITFULL; GFP, green fluorescent protein; GI, GIGANTEA; MIM, target mimicry; miRNA, microRNA; nt, nucleotide; qRT-PCR, quantitative real-time PCR; SAM, shoot apical meristem; sgRNA, single guide RNA; SMZ, *SCHLAFMUTZE*; SNZ, *SCHNARCHZAPFEN*; SOC1, *SUPPRESSOR OF OVEREXPRESSION OF CO 1*; SPL, *SQUAMOSA PROMOTER BINDING PROTEIN-LIKE*; STTM, short tandem target mimicry; SVP, SHORT VEGETATIVE PHASE; T-DNA, transfer-DNA; TOE1, *TARGET OF EAT1*; ts4, *tasselseed4*; TUB, *β-TUBULIN*-2; WT, wild type.

only 2 relatively small families, *MIR164* (i.e., *MIR164A-C*) and *MIR159* (i.e., *MIR159A-B*) [11,12], in Arabidopsis. These 2 studies showed that, similar to protein-coding genes, the *MIRNA* genes in the same family are functionally redundant. For instance, the *mir159ab* double mutant displays pleiotropic morphological defects that include curled leaves, short stature, and shorter siliques, whereas the *mir159a* and *mir159b* single mutants are phenotypically normal [12].

Target mimicry technologies, such as target mimicry (MIM) and short tandem target mimicry (STTM), have provided effective tools to silence miRNA gene families with more than 3 members by blocking endogenous mature miRNA activity [13–17]. Although generally effective, MIM and STTM apparently have their limitations. For example, residual miRNA is detectable in transgenic MIM plants, suggesting that overexpression of a target mimic is not sufficient to silence the miRNA completely. In addition, because they target the mature miRNA, both MIM and STTM fail to distinguish the functional diversity of each *MIRNA* gene. Over the past decade, the development of genome editing technology based on the clustered regularly interspaced short palindromic repeats (CRISPR)-Cas9 system has greatly advanced our ability to manipulate specific genome sequences in plants [18–22]. In particular, this method can be used to efficiently construct loss-of-function (null allele) mutants for gene families with multiple members and small gene sizes [23].

In Arabidopsis, miR172 targets a group of transcription factor genes including *APETALA2* (*AP2*), *TARGET OF EAT1* (*TOE1*), *TOE2*, *TOE3*, *SCHLAFMUTZE* (*SMZ*), and *SCHNARCH-ZAPFEN* (*SNZ*) [24,25]. Previous studies have shown that miR172 and its targets play critical roles in plant developmental transitions [26–29]. miR172 acts downstream of miR156, a miRNA in which the level gradually decreases with time after seed germination [30,31]. Accordingly, miR172 shows a temporal expression pattern that is the opposite of that of miR156 [32]. The increasing level of miR172 promotes the appearance of adult traits including the formation of trichomes (leaf hairs) on the abaxial leaf surfaces [33]. In addition, high amounts of miR172 contribute to the acquisition of floral competence [34–36]. Overexpression of miR172 leads to early flowering [32,37,38], whereas increased levels of its targets, mRNAs from *SMZ* or *SNZ*, results in a late flowering phenotype [39]. Genome-wide identification of SMZ and AP2 targets reveals that miR172-targeted AP2-like transcription factors repress flowering through inactivation of the florigen gene *FLOWERING LOCUS T* (*FT*) in leaves and MADS-box genes such as *APETALA1* (*AP1*) and *SUPPRESSOR OF OVEREXPRESSION OF CO 1* (*SOC1*) at the shoot apices [39,40].

It has been shown that miR172 also plays a regulatory role in floral patterning in Arabidopsis. *AP2* and miR172 show a complementary expression manner in developing floral organs: *AP2* is predominantly expressed in the outer floral whorls, while miR172 accumulates to high levels in the centers of the floral primordia [41,42]. The expression of *AP2* and miR172 partially overlaps at the boundary between the perianth and the reproductive organs in the third whorl. Reducing miR172 activity by overexpressing *MIM172* in the third whorl converts stamens into petals [42].

miR172 belongs to one of the miRNA gene families that are ubiquitous and generally highly expressed across terrestrial plant species [43–46]. Careful sequence analysis has shown that miR172 is not present in either the bryophyte *Physcomitrella patens* or the lycophyte *Selaginella moellendorffii* [47], suggesting that the origin of miR172 may play a critical function in the evolution of the vascular plants. However, it is currently unknown whether miR172 is absolutely required for viability in higher plants such as Arabidopsis and rice. In addition, the functional redundancy and specificities among *MIR172* genes remain to be clarified. Here, we report the functional investigation of the *MIR172* gene family in Arabidopsis using CRISPR-Cas9 technology. We find that the 5 *MIR172* gene family members exert functional specificities

in regulating diverse aspects of plant developmental processes. The detailed analysis of the role of miR172 in the floral transition suggests that the specificities of *MIR172* genes can be explained by their distinct expression patterns and different responsiveness to plant age, ambient temperature, and photoperiod.

## Results

### Generation of *MIR172* knockout mutants by CRISPR-Cas9 technology

In the Arabidopsis genome, the miR172 gene family consists of 5 members, *MIR172A* to *MIR172E* (Fig 1A) [25,48]. Because of the small sizes of these genes (<200 base pairs) and their noncoding property, loss-of-function mutants of *MIR172* family members in the same genetic background are rarely obtained by canonical mutagenesis methods such as ethyl methanesulfonate treatment, fast neutron irradiation, or T-DNA tagging. To circumvent this problem, we generated *mir172* mutants in the Columbia-0 (Col-0) accession of *A*. *thaliana* using CRISPR-Cas9 technology with the egg cell-specific promoter [49]. To generate null alleles, we sought to create mutants with large fragment deletions by designing 2 single guide RNAs (sgRNAs) within or flanking the stem-loop region (Fig 1B). Using PCR genotyping, we successfully identified plants carrying mutations in every *MIR172* family member in the T$_1$ generation (Fig 1B; S1 Fig). For the *MIR172A* and *MIR172C* loci, the entire stem-loop regions were deleted; for the *MIR172B* and *MIR172D* loci, the deletions occurred in the loops of the stem-loop regions; for the *MIR172E* locus, the sequence corresponding to the miRNA* (passenger strand) was deleted.

To confirm that every mutation gave a null allele, the mutated *MIR172* genes were cloned and overexpressed individually in wild-type (WT) plants using the constitutive CaMV *35S* promoter. Overexpression of the WT *MIR172A* led to an early flowering phenotype under long days [32,41,50], whereas the overexpression of the mutated versions of the *MIR172A*, *MIR172C*, and *MIR172E* genes did not alter flowering time (Fig 1C and 1D; S1 Table). In agreement with the nature of mutations (i.e., deletions in the loops of the stem-loop regions), the transgenic plants with high levels of *mir172b* and *mir172d* flowered slightly earlier than WT (Fig 1C and 1D; S1 Table). Therefore, we conclude that the *mir172a*, *mir172c*, and *mir172e* mutant genes are completely null. In contrast, residual miR172 may exist in the *mir172b* and *mir172d* mutants. The mutants were then backcrossed to WT to remove the transgene and off-target mutations. In total, 24 *mir172* multiple-mutant lines in different combinations were generated by crossing and PCR-based genotyping (S2 Table).

We examined mature miR172 levels by quantitative real-time PCR (qRT-PCR). As shown in S2A Fig, miR172 accumulates at low levels in the seedling stage and gradually increases with development under long days. The highest expression level was observed in inflorescences (S2A Fig). miR172 was barely detected in the *mir172* quintuple mutant (S2B Fig). The comparison of miR172 abundance between WT and the *mir172* mutants showed that *MIR172A* and *MIR172B* contribute to most of the mature miR172 pool in 12-day-old plants in long days (S2B Fig). In contrast, mutation in *MIR172C* led to a slight decrease in miR172. The *mir172d* and *mir172e* mutants accumulated the same level of miR172 as the WT. Consistent with the notion that miR172 represses its targets mainly through translational inhibition [41], we did not find significant change in abundance of miR172 targets in the *mir172* mutants (S2C to S2L Fig). Thus, these results confirm that miR172 expression is nearly abolished in the *mir172* quintuple mutants.

### Phenotypic analyses of *mir172* multiple-mutant lines

As described earlier, although the target mimicry approach can be effective, miR172 is still detectable in the transgenic *MIM172* plants [14]. To understand whether miR172 is absolutely

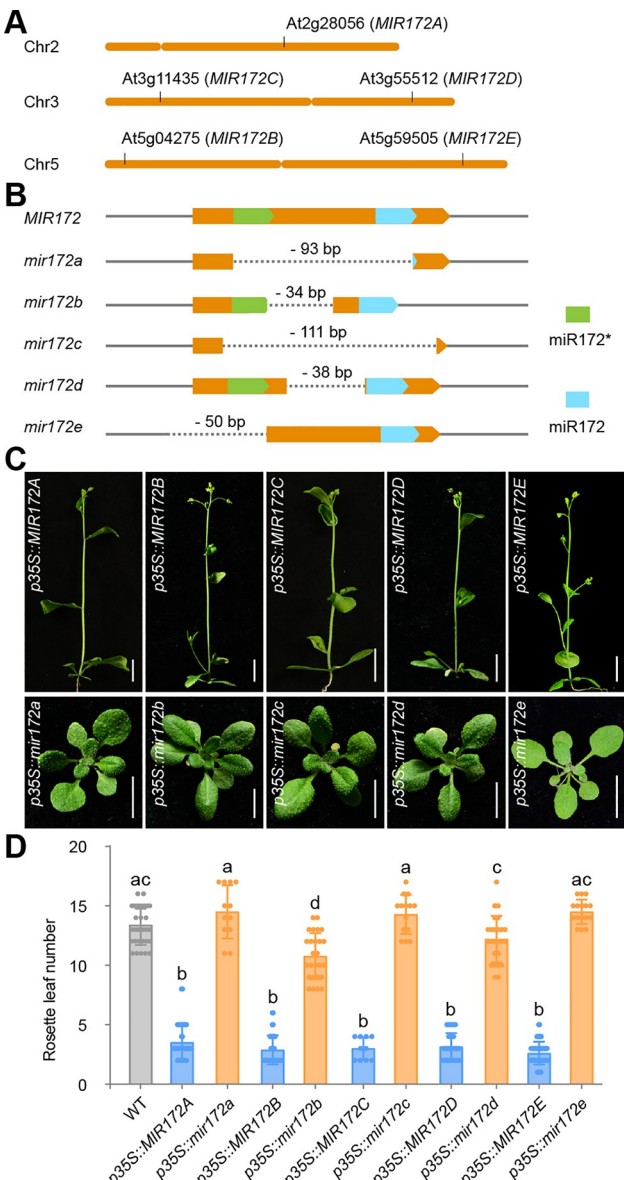

**Fig 1. Generation of *MIR172* mutants by CRISPR-Cas9 technology.** (A) Location of the *MIR172* genes on Arabidopsis chromosomes. The short and long arms of chromosomes 2, 3, and 5 are shown. (B) The genomic structure of the *MIR172* gene loci in WT and the *mir172* mutants. Orange, stem-loop of *MIR172* gene; green, miRNA*; blue, miRNA. (C) Validation of the *mir172* mutants. The WT or mutated versions of *MIR172* genes were cloned and overexpressed in WT. Please note that overexpression of the WT version of *MIR172* led to early flowering time phenotype. Scale bars represent 1 cm. (D) Flowering time measurement. The number of rosette leaves was counted. Plants were grown at 22˚C in long days. The statistically significant differences are determined by ordinary one-way ANOVA ($p < 0.05$). See also S4 Table. The data underlying this figure are included in S1 Data. CRISPR, clustered regularly interspaced short palindromic repeats; miRNA, microRNA; sgRNA, single guide RNA; WT, wild type.

required for plant growth and development, we performed phenotypic analyses of the *mir172* quintuple mutants. The *mir172* quintuple mutants are viable when grown at 22˚C in the growth chamber under both long-day and short-day conditions. The quintuple mutants could also survive and produce a large number of seeds when grown outdoors (S3 Fig). This result indicates that, despite its ancient origin, miR172 is not essential for the completion of the seed-to-seed life cycle in Arabidopsis.

Compared to the WT, the *mir172* quintuple mutants displayed pleiotropic phenotypes. During the vegetative phase, the development of abaxial trichomes was delayed in the *mir172* quintuple mutants under long days (Fig 2A; S4 Fig), which is consistent with previous observations [33]. The contributions of each gene to abaxial trichome initiation were not equal: The

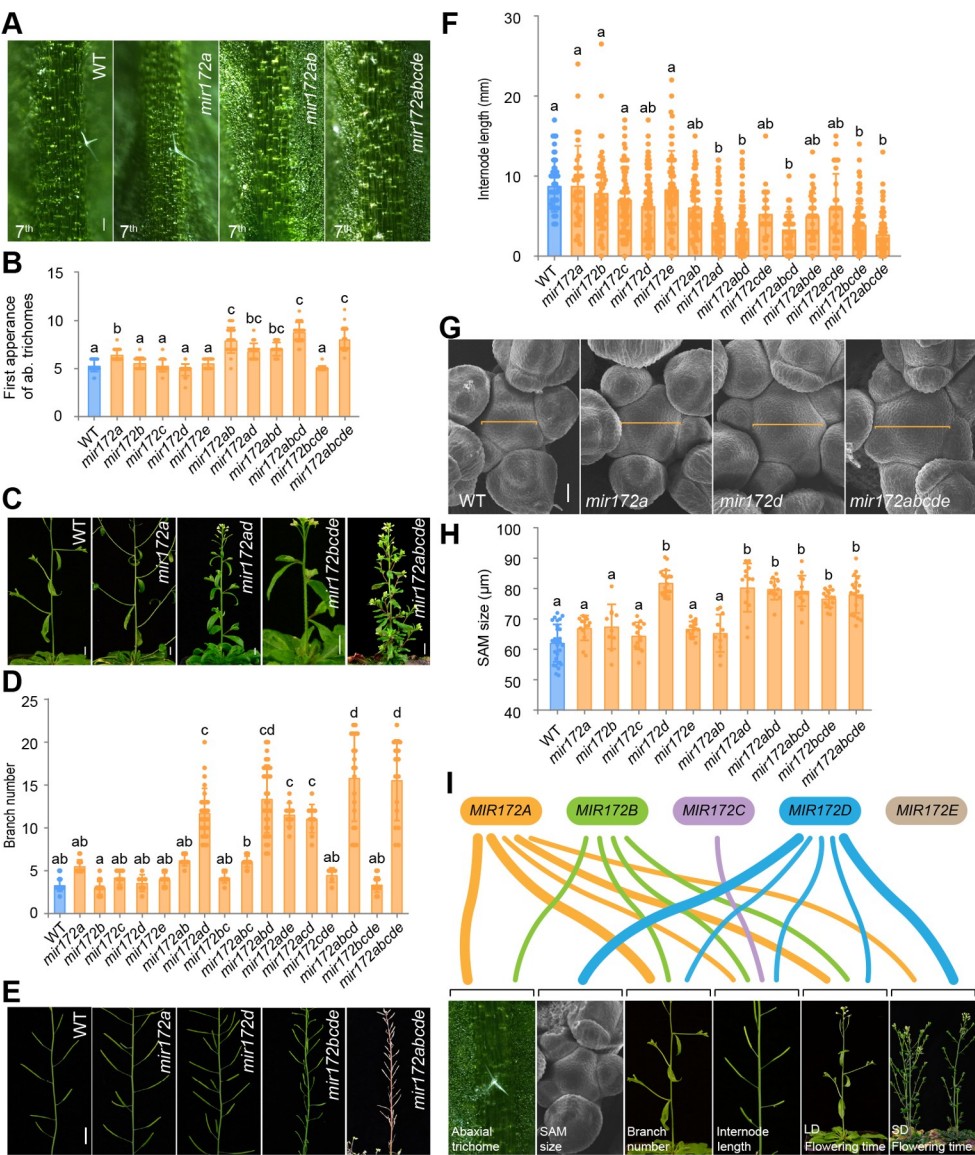

**Fig 2. Phenotypic analyses of *mir172* mutants.** (A) Abaxial trichomes phenotype. The abaxial surface of the 7th leaf was shown. Scale bars represent 200 μm. (B) Quantification of the abaxial trichomes in different genotypes. The 1st leaf with abaxial trichomes was scored. Error bars represent SEM (*n* = 18–28). (C) Shoot branch phenotype. Scale bars represent 1.0 cm. (D) Quantification of shoot branch numbers in different genotypes. Error bars represent SEM (*n* = 6–37). The statistically significant differences are determined by ordinary one-way ANOVA (*p* < 0.05). (E) Internode length phenotype. Scale bars represent 1.0 cm. (F) Quantification of internode length. Error bars represent SEM (*n* = 27–153). The statistically significant differences are determined by ordinary one-way ANOVA (*p* < 0.05). (G) Inflorescence SAM size. Orange line, diameter of inflorescence SAM. Scale bars represent 20 μm. (H) Quantification of the size of inflorescence SAM. Error bars represent SEM (*n* = 10–29). The statistically significant differences are determined by ordinary one-way ANOVA (*p* < 0.05). (I) Sankey diagram showing the contribution of each *MIR172* gene to diverse developmental processes. The individual *MIR172* gene was shown in different color. Line thickness stands for the contribution of each gene. The data underlying this figure are included in S1 Data. LD, long days; SAM, shoot apical meristem; SD, short days; WT, wild type.

*mir172ab* double mutant exhibited the same early trichome phenotype as the *mir172* quintuple mutant (Fig 2A and 2B), whereas mutations in the other *MIR172* genes did not affect the timing of trichome initiation. Thus, these results suggest that *MIR172A* and *MIR172B* play dominant roles in the timing of trichome initiation (Fig 2I).

The *mir172* quintuple mutants exhibited striking phenotypes after flowering. The number of branches on the primary bolt was markedly increased (Fig 2C and 2D), accompanied by shortened internodes (Fig 2E and 2F). Interestingly, we found that the meristem size in the quintuple mutant was increased (Fig 2G to 2I). This phenotype is in a good agreement with a recent report that a MADS-box transcription factor FRUITFULL (FUL) regulates the timing of shoot apical meristem (SAM) termination through miR172-targeted AP2-like genes [51]. The comparisons among multiple-mutant lines further revealed that *MIR172D* plays a key role in regulating SAM size, whereas all of the *MIR172* genes except *MIR172E* cooperatively control internode elongation (Fig 2I). Branch number is mainly regulated by *MIR172A* and *MIR172D*. Therefore, our results indicate that the 5 *MIR172* gene family members are functionally redundant but exert individual specificities in regulating diverse aspects of plant development (Fig 2I).

It has been proposed that miR172 regulates floral patterning by modulating *AP2* expression at the third whorl [35,42,52–54]. Indeed, we found that the first few flowers on the primary bolt showed homeotic transformation (S5A Fig). However, this defect disappeared quickly with the development of the inflorescence (S5B Fig), implying that miR172 may play a minor role in floral patterning during the late reproductive stages under growth chamber conditions.

## Flowering time analyses of *mir172* mutants under long-day conditions

A previous study showed that down-regulation of miR172 activity by overexpression of a target mimic (*35S::MIM172*) leads to a late flowering phenotype [14]. However, it remains unclear whether all the *MIR172* family members are involved in flowering time regulation and, if so, whether the different members contribute differently to this process. To answer these questions, we scored the flowering times of the single and higher-order *mir172* mutant plants grown under both long and short days. As shown in Fig 3B and S1 Table, the *mir172a* mutant, but not other *mir172* single mutants, exhibited a late flowering phenotype in long days. Consistently, the *mir172bcde* quadruple mutant flowered at the same time as WT (Fig 3A and 3B; S1 Table). Moreover, the double or triple mutant combinations that included *mir172a* flowered significantly later than WT, whereas the combinations without the *mir172a* mutation flowered normally, like WT plants (Fig 3A and 3B; S1 Table). *MIR172E* appears to play only a minor role in regulating flowering time because the *mir172abcd* quadruple mutant had the same flowering time as the *mir172* quintuple mutant. Both the *mir172abcd* and *mir172abde* quadruple mutants flowered later than *mir172acde*, suggesting that *MIR172B* makes a modest contribution to the floral transition. Taken together, we conclude that the contribution of each *MIR172* gene family member to flowering time under long days is: *MIR172A*>*MIR172B*>*MIR172D*>*MIR172C*>*MIR172E* (Fig 2I).

It has been shown that the level of miR172 increases with age, and this promotes the gain of reproductive competence [33]. qRT-PCR assays showed that the gradual increase in miR172 levels was largely compromised in the *mir172ab* double mutant (Fig 3C). Induction of *FT* by the photoperiod pathway plays a critical role in flowering in long days [55]. The *mir172ab* double mutant plants had lower levels of *FT* than did WT plants in long days (Fig 3D), indicating that these 2 *MIR172* members alleviate the repression of *FT* expression by miR172-targeted AP2s in leaves.

## Expression pattern of *MIR172* genes in plants grown under long days

We generated the fluorescent protein [green fluorescent protein (GFP) or Venus]-based reporters for the 5 *MIR172* gene family members. To ensure that the promoter regions covered all of the

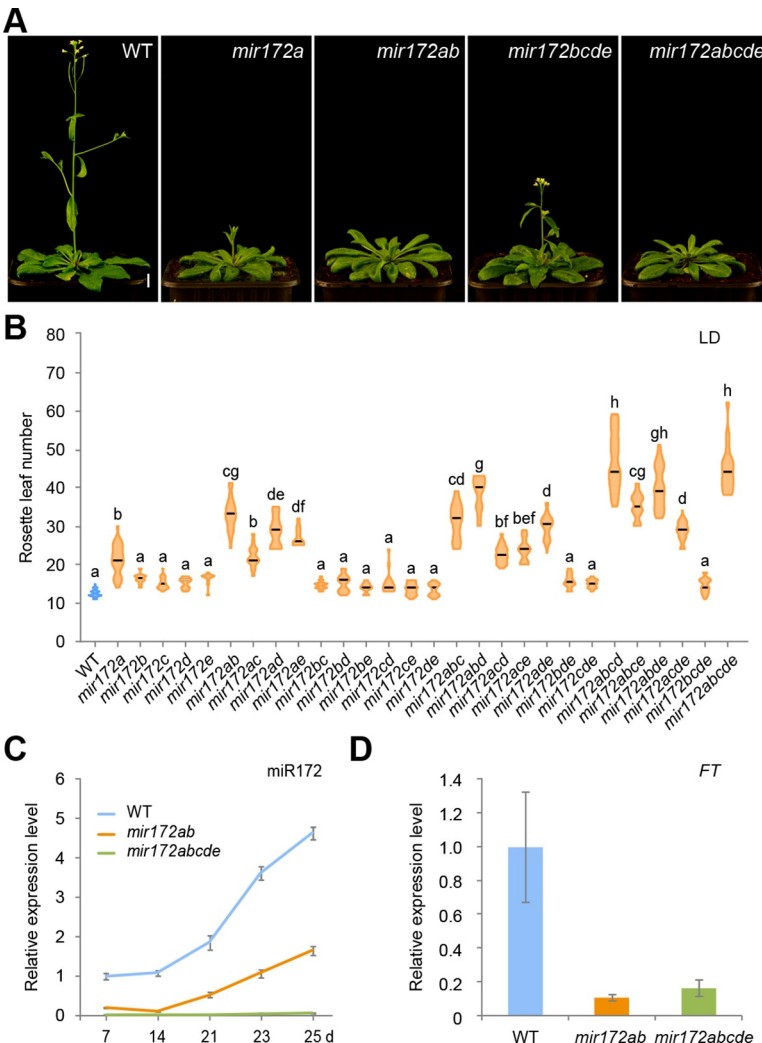

**Fig 3. Flowering time of WT and the *mir172* mutants in long days.** (A) Flowering time phenotype of the *mir172* mutants. Plants were grown at 22°C in long days. One representative plant is shown. (B) Quantification of flowering times of the *mir172* mutants. Lines show mean (*n* = 7–36). The statistically significant differences are determined by ordinary one-way ANOVA (*p* < 0.05). See also S1 and S4 Tables. (C) Expression of miR172 in WT, *mir172ab*, and *mir172abcde* mutants. Plants were grown at 22°C in long days. The shoot apices and developing leaves were used for qRT-PCR analysis. Expression was normalized to *TUB*. Two technical replicates for each biological replicate (*n* = 2) were performed. Error bars represent SD. (D) Expression of *FT* in WT and the *mir172* mutants. The leaves of 15-day-old plants grown at 22°C in long days were harvested at ZT16 and used for qRT-PCR analyses. Expression was normalized to *TUB*. Two technical replicates for each biological replicate (*n* = 3) were performed. Error bars represent SD. The data underlying this figure are included in S1 Data. *FT*, *FLOWERING LOCUS T*; LD, long days; qRT-PCR, quantitative real-time PCR; *TUB*, *β-TUBULIN-2*; WT, wild type; ZT16, Zeitgeber time 16.

regulatory sequences, we surveyed our transposase-accessible chromatin sequencing (ATAC-seq) datasets, which reveal the chromatin accessibility at a given gene locus [56]. The promoter sequences that contained all of the accessible regions in the intergenic region were cloned and placed upstream of the coding regions of *GFP* or *Venus* (S6 Fig). For each construct, we examined over 20 individual T$_1$-generation lines that gave consistent and reliable expression patterns. We chose 1 representative T$_2$ line for the subsequent analyses. We did not detect the GFP fluorescent signals for *MIR172E* under long days (S7 Fig), suggesting that *MIR172E* is weakly expressed and is not required for development when the plants are grown under LD conditions.

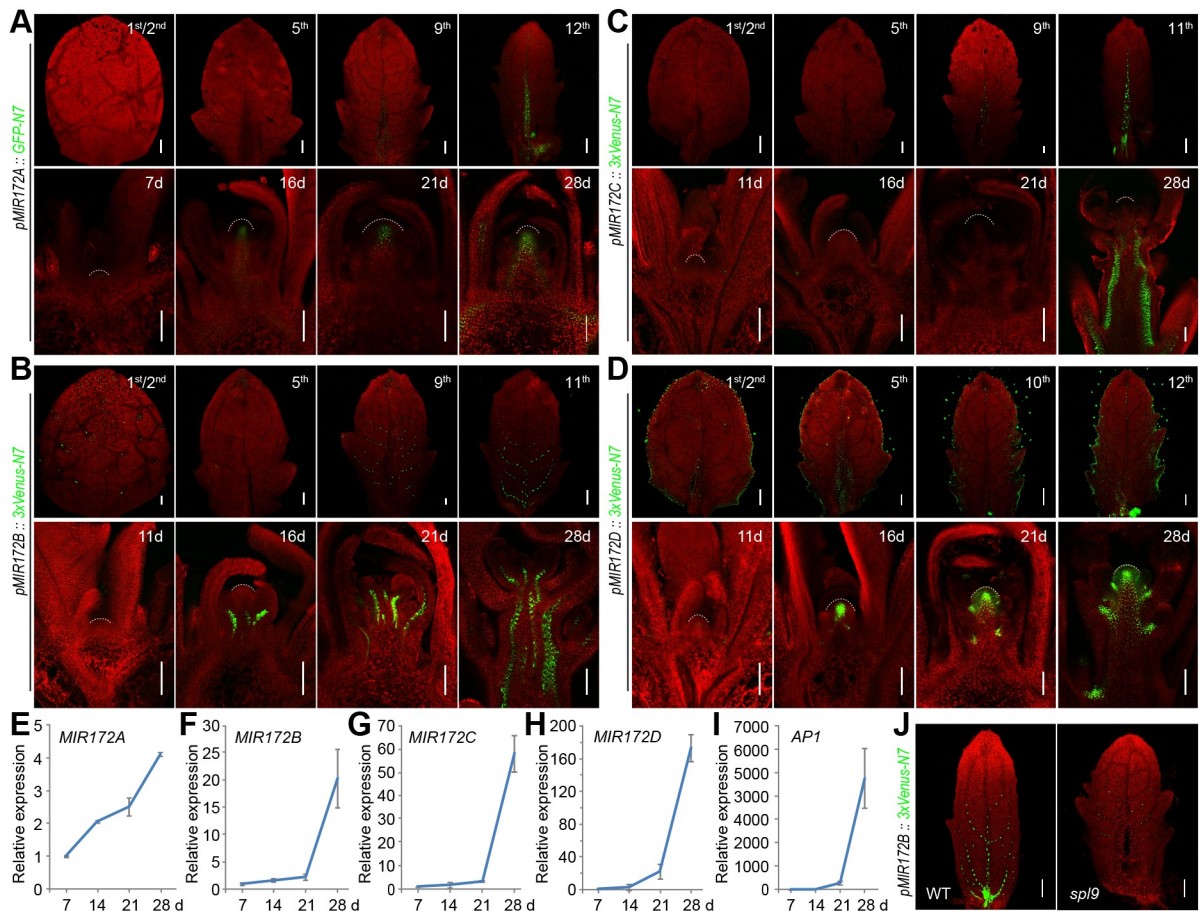

**Fig 4. Expression pattern of *MIR172* genes in long days.** (A–D) Analyses of *MIR172* reporters. Plants were grown at 22°C in long days. The leaves (upper panels) and shoot apices (lower panels) were examined. Please note that the plants start to flower 3 weeks (approximately 21 days) after seed germination. Over 20 T$_1$ independent lines for each reporter were examined, and the results of 1 representative T$_2$ line are shown. Dash line marks the SAM. Scale bars represent 100 μm. (E–I) Expression of *MIR172* genes (E–H) and *AP1* (I) in WT. *AP1* was monitored as an indication for floral transition. Plants were grown at 22°C in long days and harvested at different time points. Expression was normalized to *TUB*. Two technical replicates for each biological replicate (*n* = 2) were performed. Error bars represent SD. (J) Expression of the *MIR172B* reporter in WT and the *spl9* mutant. One representative leaf of the same developmental age is shown. Scale bars represent 100 μm. The same confocal settings were used for scanning for each reporter line (A–D and I). The data underlying this figure are included in S1 Data. *AP1*, *APETALA1*; SAM, shoot apical meristem; *TUB*, *β-TUBULIN-2*; WT, wild type.

We next focused on the expression of the *MIR172* reporter genes in the leaf vascular tissues or the SAM, where the floral induction occurs. In long days, *MIR172B* and *MIR172C* were expressed at low levels in seedlings and became detectable in the vascular tissues when the plants entered the adult phase and eventually flowered (Fig 4B and 4C). *MIR172A* is actively transcribed in both vascular tissue and the SAM, and its promoter strength increased as development progressed (Fig 4A). *MIR172D* exhibited a different expression pattern in which the Venus fluorescence was constitutively observed at the margins of Arabidopsis leaves (Fig 4D). A high level of expression of *MIR172D* was evident in the SAM of 16-day-old plants. The temporal expression pattern of these reporters was verified by qRT-PCR and quantification of GFP or Venus densities (Fig 4E to 4I; S8 Fig). Overall, these analyses are consistent with the abovementioned genetic analysis data where *MIR172A* and *MIR172B* play predominant roles in flowering in long days. The activation of these 2 genes in the leaf vascular tissues primes the activation of *FT* by the photoperiod pathway.

Genetic analyses have placed miR172 downstream of miR156-targeted *SQUAMOSA PRO-MOTER BINDING PROTEIN-LIKE* (*SPL*) genes [57–60]. The gradual decrease in miR156 levels with increasing plant age leads to the up-regulation of *SPL* genes, which subsequently activates miR172 [33,61]. In situ hybridization assays have shown that *SPL9* is expressed in leaf anlagen and the vascular tissues [30,62]. Compared to WT, the promoter activities of *MIR172B* and *MIR172C* were moderately reduced in the leaves of *spl9* mutant plants (Fig 4J; S9B and S9D Fig), suggesting that SPL9 contributes to the increased level of miR172 in the leaves of adult plants through these 2 *MIR172* gene family members. The transcriptional activity of *MIR172A* was not altered in the *spl9* mutant, suggesting that one or more other miR156-targeted SPLs may regulate its expression (S9A Fig). Indeed, our recent work has shown that the *SPL2*, *SPL10*, and *SPL11* genes are also highly expressed in leaf vascular tissues [63].

## *MIR172A* and *MIR172D* play dominant roles in determining flowering time under short days

In short days, we found that *MIR172D* played a critical role in flowering because the *mir172d* single mutant exhibited an obvious late flowering phenotype (Fig 5A and 5B; S1 Table). This phenotype was further enhanced by mutation of *MIR172A*. The *mir172ad* double mutant plants flowered nearly as late as the *mir172* quintuple mutant plants (Fig 5B; S1 Table). In contrast, the flowering times of the other double mutants was comparable to that of WT. It is well documented that the activation of a subset of MADS-box genes and *LFAFY* in the SAM evokes flowering in short days [55,64]. In situ hybridization showed that miR172 abundance is markedly reduced in the shoot apices of the *mir172ad* double mutant (Fig 5C). Expression analyses further revealed that the up-regulation of MADS-box genes such as *FUL* and *SOC1* is delayed in the shoot apices of *mir172ad* mutant (Fig 5D). Therefore, we conclude that the relative contribution of each *MIR172* gene family member to flowering time in short days is: *MIR172D*>*MIR172A*>*MIR172B* = *MIR172C*>*MIR172E* (Fig 2I).

We next analyzed the *MIR172* reporter activities in plants grown in short days. Expression of *MIR172B* and *MIR172C* was barely detectable in the vegetative phase (Fig 6B and 6C; S10 Fig). Consistent with the role of *MIR172A* and *MIR172D* in flowering under short days, both reporters were active in the SAM, with the promoter strength increasing as development progressed (Fig 6A and 6D; S10 Fig). Among the miR156-targeted SPL genes, *SPL15* is predominantly expressed in the SAM and coordinates the basal floral promotion pathways required for flowering under noninductive short-day conditions [61]. Consistent with this notion, the expression of *MIR172D* was greatly reduced in the SAM of the *spl15* mutant (Fig 6F; S9C and S9D Fig). On the contrary, *MIR172A* expression was largely unaffected by the mutation in *SPL15* (S9A and S9D Fig).

In summary, the flowering times and expression analyses described above suggest that *MIR172A*, *MIR172B*, and *MIR172D* are the primary regulators of flowering time in the *MIR172* gene family. The spatially localized SPL-*MIR172* pairs promote the acquisition of floral competence in different tissues under different growth conditions (Fig 8A and 8B). In long days, miR172 functions in both the SAM and vascular tissues, with *MIR172A*, *MIR172B*, and *MIR172D* playing dominant roles. In the vasculature of leaves, *MIR172B* is induced by SPL9, whereas *MIR172A* is activated probably by other miR156-targeted SPLs. The up-regulation of *MIR172A* and *MIR172B* relieves the repression of *FT* by miR172-targeted genes that encode AP2-like transcription factors. Consequently, *FT* is activated in leaves by the photoperiod pathway, and the mobile protein is transported to the SAM where it induces the floral transition. The SPL15-*MIR172D* pair in the SAM plays a less important role in flowering under long-day conditions because the photoperiod pathway is dominant in Arabidopsis, and the

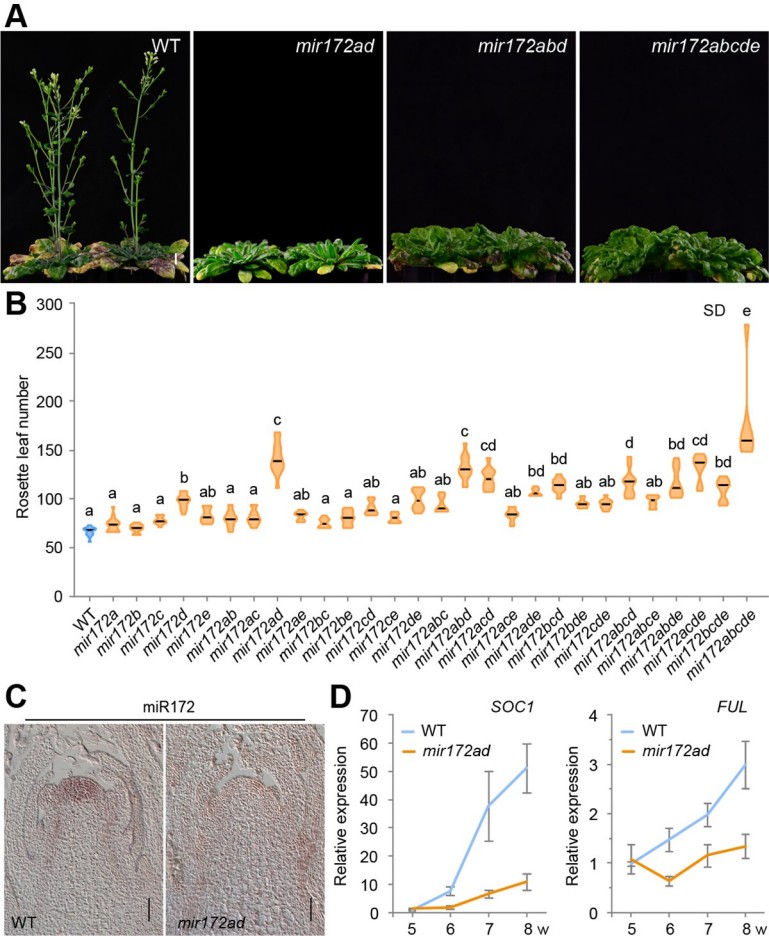

**Fig 5. Flowering time of WT and the *mir172* mutants in short days.** (A) Flowering time phenotype of the *mir172* mutants. Plants were grown at 22°C in short days (SD). Four representative plants are shown. (B) Quantification of flowering times of the *mir172* mutants. Lines show mean (*n* = 6–16). The statistically significant differences are determined by ordinary one-way ANOVA (*p* < 0.05). See also S1 and S4 Tables. (C) Expression of miR172 in the shoot apices of WT and the *mir172ad* mutant. Plants were grown at 22°C in short days. Scale bars represent 50 μm. (D) Expression of *SOC1* and *FUL* in WT and the *mir172ad* mutants. Plants were grown at 22°C in short days, and the shoot apices were harvested at different time points as indicated. Expression was normalized to *TUB*. Two technical replicates for each biological replicate (*n* = 2) were performed. Error bars represent SD. The data underlying this figure are included in S1 Data. *FUL*, *FRUITFULL*; *SOC1*, *SUPPRESSOR OF OVEREXPRESSION OF CO 1*; TUB, *β-TUBULIN-2*; WT, wild type.

repressive role of miR172-targted AP2s on floral transition in the shoot apex can be eventually bypassed by FT.

In short days, miR172 activity is crucially important for floral transition at the SAM, with *MIR172A* and *MIR172D* being the most important (Fig 8A and 8B). The gradual increase in *SPL15* levels promotes the transcription of *MIR172D* in the SAM. As a result, the accumulation of mature miR172 leads to down-regulation of miR172-targeted *AP2*-like genes, which eventually facilitates the activation of floral promoting MADS-box genes. How *MIR172A* is progressively activated in the SAM in short days is currently unknown. Possible activators include SPL2 and SPL10 which are also highly abundant in the SAM (S11 Fig). It should be noted that the contribution of *MIR172B* to floral transition in short days may be underscored because the mutant allele is not completely null (Fig 1C and 1D).

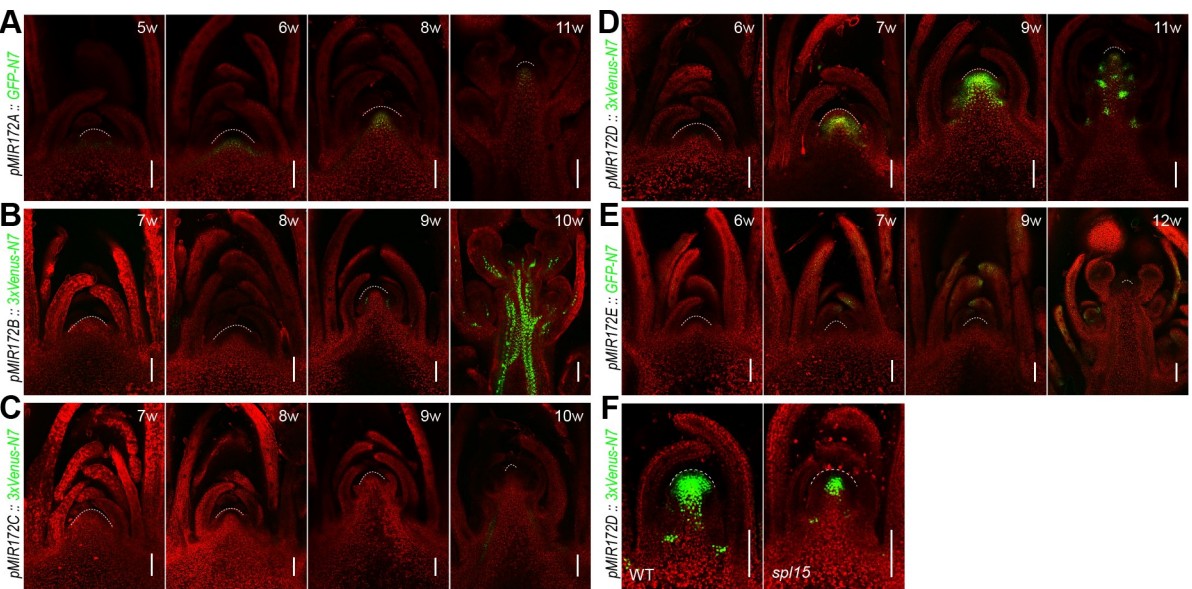

**Fig 6. Expression pattern of *MIR172* genes in short days.** (A–E) Analyses of *MIR172* reporters. Plants were grown at 22°C in short days. The shoot apices were examined. Please note that the plants start to flower after 10 weeks. Over 20 T$_1$ independent lines for each reporter were examined, and the results of 1 representative T$_2$ line are shown. Dash line marks the SAM. Scale bars represent 100 μm. (F) Expression of *MIR172D* reporter in WT and the *spl15* mutant. Dash line marks the SAM. Scale bars represent 100 μm. The same confocal settings were used for scanning for each reporter line. SAM, shoot apical meristem; WT, wild type.

### The role of *MIR172* genes in response to ambient temperature

Previous studies have shown that overexpression of miR172 leads to an early flowering phenotype at both 16 and 23°C [65,66]. Moreover, miR172 levels are higher at 23°C than at 16°C, probably due to an enhancement in the processing of the primary miR172 transcripts mediated by the Arabidopsis RNA-binding protein FLOWERING CONTROL LOCUS A (FCA) [66]. However, due to the extremely early flowering phenotype of the miR172-overexpression line, it remains unclear whether miR172 is indeed required for thermosensory flowering. We found that the flowering of both WT and *mir172* quintuple mutant plants was accelerated at 28°C (Fig 7A), indicating that elevated ambient temperature promotes flowering independent of miR172. In contrast, although the flowering of the WT plants was delayed in 16°C, the *mir172* quintuple mutants started to bolt with nearly the same number of rosette leaves at 16 and 22°C (Fig 7B and 7C). Thus, this result indicates that low ambient temperature regulates flowering largely through miR172.

To understand which *MIR172* genes are responsible for the temperature response, we compared the flowering times of the single and higher-order *mir172* mutants grown at different temperatures. As shown in Fig 7B, all single *mir172* mutants showed moderate reduction in thermosensory flowering response with *MIR172A* likely playing a major role within the gene family (Fig 7B and 7C). The double-mutant plants harboring the mutation in *MIR172A* (*mir172ab*, *mir172ac*, and *mir172ad*) flowered at nearly the same time when grown at 16°C compared to 22°C (Fig 7B and 7C).

We next explored the expression of individual *MIR172* genes in response to low ambient temperature. Consistent with previous results [65,66], qRT-PCR assays revealed that the abundance of miR172 was elevated in plants grown at 22°C, as compared to 16°C (Fig 7D). Among the reporters examined, we found that *MIR172C* and *MIR172D* responded to the temperature change (Fig 7E; S12 Fig). The expression of *MIR172C* was greatly increased in the vasculature

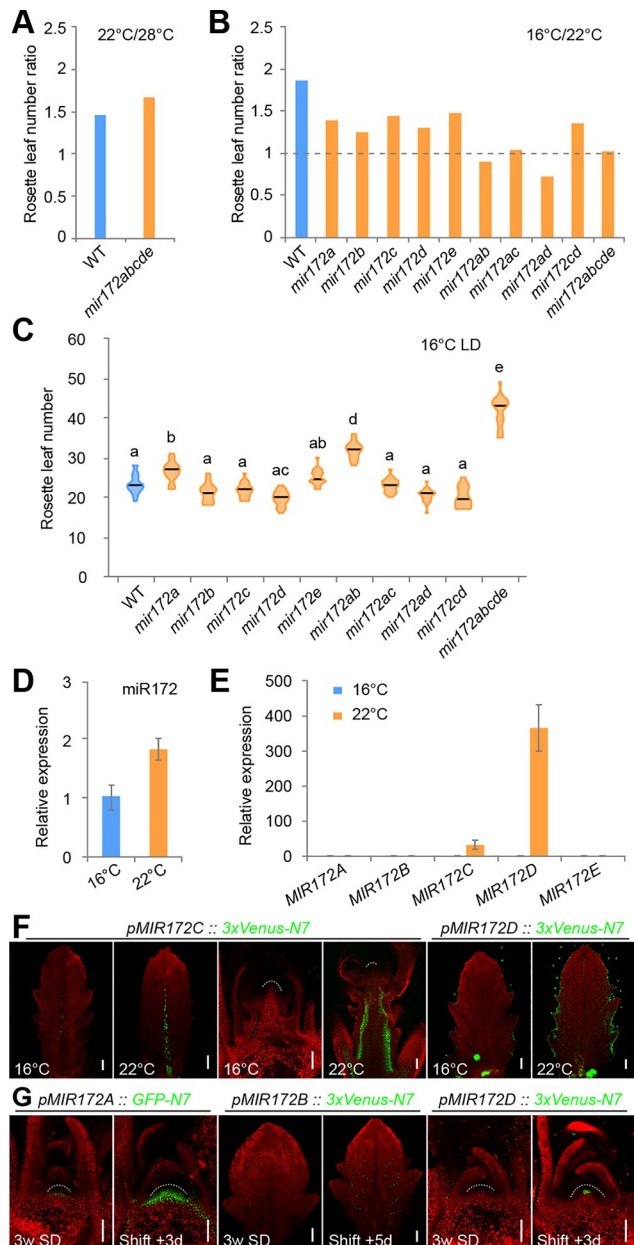

**Fig 7. Expression of *MIR172* genes in response to environment cues.** (A) Comparison of the flowering time of WT and the *mir172abcde* mutants grown at 22˚C and 28˚C in long days. Rosette number ratio was calculated by the mean value at 22˚C divided by that at 28˚C. (B) Comparison of the flowering time of WT and the *mir172* mutants grown at 16˚C and 22˚C in long days. Rosette number ratio was determined by the mean value at 16˚C divided by that at 22˚C. (C) Quantification of flowering time phenotype of the *mir172* mutants at 16˚C in long days. Lines show mean (*n* = 18–24). The statistically significant differences are determined by ordinary one-way ANOVA (*p* < 0.05). See also S1 and S4 Tables. (D and E) Expression of miR172 (D) and *MIR172* genes (E) at 16˚C and 22˚C in long days. Expression was normalized to *TUB*. Two technical replicates for each biological replicate (*n* = 3) were performed. Error bars represent SD. (F) Analyses of *MIR172C* and *MIR172D* reporters in the shoot apices. Plants were grown at 16˚C in long days. (G) Expression of *MIR172A*, *MIR172B*, and *MIR172D* reporters in response to photoperiod. The plants were grown in short days (SD) for 3 weeks and shifted to long days (shift + 3 d or 5 d). The same confocal settings were used for scanning for each reporter line (F and G). Dash lines mark the SAM. See also S13 Fig. The data underlying this figure are included in S1 Data. LD, long days; SAM, shoot apical meristem; SD, short days; TUB, *β-TUBULIN-2*; WT, wild type.

of leaves and the inflorescence stem, whereas the promoter activity of *MIR172D* was enhanced with the increase in temperature, mainly in the vascular tissues and leaf epidermis (Fig 7F). Altogether, these results indicate that, in addition to modulating miR172 abundance at the posttranscriptional level as previously proposed [66], low ambient temperature may control flowering by regulating the expression of *MIR172C* and *MIR172D* at the transcriptional level (Fig 8A).

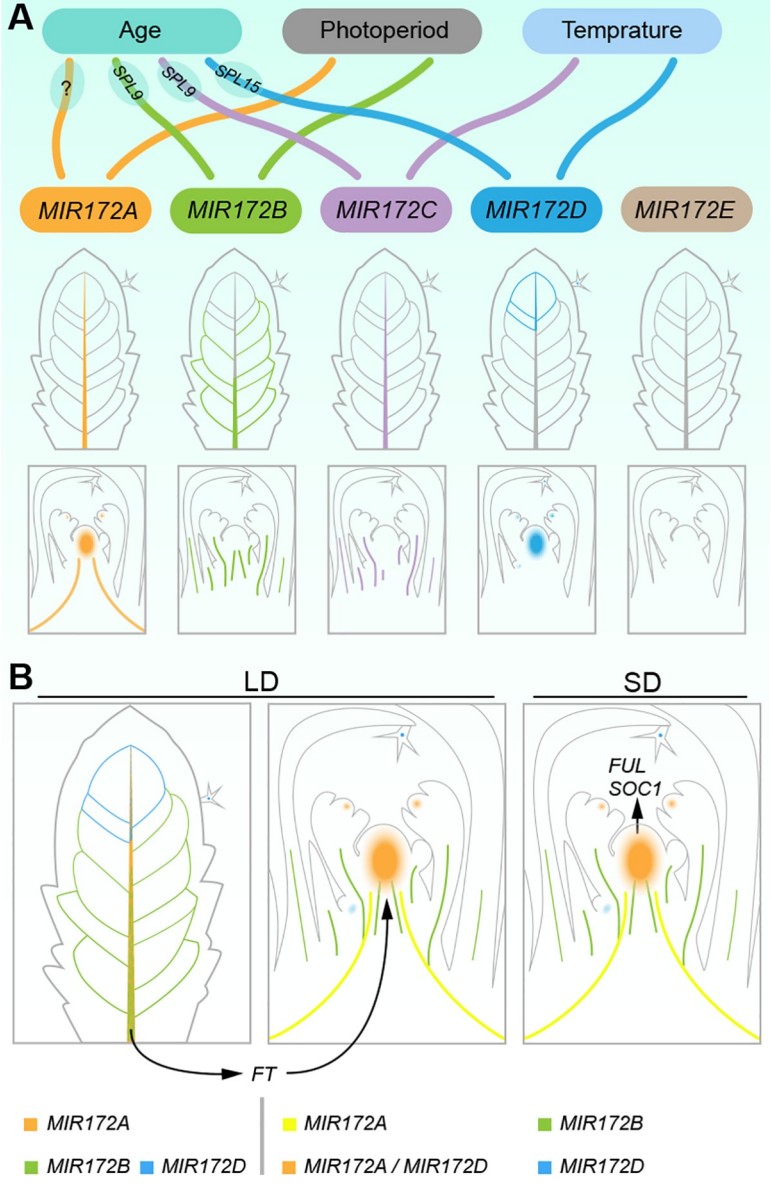

**Fig 8. Distinct expression pattern and different responsiveness of *MIR172* genes to plant age, ambient temperature, and photoperiod.** (A) Schematic of the expression pattern of *MIR172* genes. Individual *MIR172* gene is shown in different color. Two *SPL/MIR172* pairs (SPL9-*MIR172B/C* and SPL15-*MIR172D*) contribute to the acquisition of floral competence in leaf vascular tissue and shoot apex, respectively. (B) Mode of *MIR172* genes in regulating flowering time in LD and SD. Different combinations of *MIR172* genes are shown in different colors. In LD, the products of *FT* move from leaves to shoot apex (black lines). *FT, FLOWERING LOCUS T*; *FUL, FRUITFULL*; LD, long days; SD, short days; *SOC1, SUPPRESSOR OF OVEREXPRESSION OF CO 1*.

### The expression of *MIR172* genes in response to photoperiod

Finally, we explored whether photoperiod, another exogenous floral inductive cue, can affect the transcription of *MIR172* genes. To this end, we grew the transgenic plants expressing the *MIR172* reporters in short days for 3 weeks and then shifted them to long days. As shown in Fig 7G and S13 Fig, we did not detect strong induction of all the reporters 1 day after the plants were shifted to long-day conditions. At day 3, the transcriptional activities of *MIR172A* and *MIR172D* were strongly induced in the shoot apex, whereas increases in *MIR172A* and *MIR172B* expression were observed in the leaf vascular tissues (Fig 7G; S13 Fig). The expression of *MIR172C* did not respond to long days (S13 Fig). We did not find ectopic expression of *Venus* or *GFP*, suggesting that long-day conditions only regulate the amplitude of *MIR172* expression. Because the expression of floral-promoting genes such as *FUL* and *SOC1* are rapidly induced in the SAM 1 day after the shift [30], the increased levels of *MIR172A* and *MIR172D* in the SAM may be caused by a cell fate transition rather than by a direct photoperiod response. Therefore, photoperiod modulates miR172 abundance at least at two different levels: transcriptional activation of *MIR172A* and *MIR172B* in the vasculature of leaves (Fig 8A) and promoting the processing of miR172 precursors through the plant circadian oscillation regulator GIGANTEA (GI) [67].

## Discussion

Evolutionarily conserved miRNAs are usually encoded by multiple gene families. The comprehensive characterization of the function of each of the gene family members is difficult, largely due to a lack of null knockout mutants. With the development of CRISPR-Cas9 technology, it is now possible to generate the corresponding mutant alleles and investigate their biological functions. More importantly, the generation of higher-order mutants carrying different combinations of mutations in each miRNA gene family member allows us to explore the function and contribution of each member. In this study, we have demonstrated that the Arabidopsis *MIR172* gene family members are not only redundant but also confer functional specification (Fig 2I). Our reporter analyses further reveal that this specificity can be attributed to their distinct expression pattern, in addition to variations in miRNA processing efficiency. For example, consistent with its being highly expressed in the SAM (Fig 4D), *MIR172D* is the only family member that regulates SAM size. The promote activity of *MIR172D* could also be detected in other tissues such as lateral shoots and vascular bundles in the stem (Fig 4D), where it participates in the control of shoot branching and internode elongation. Similar result has been reported for the Arabidopsis *MIR164* family, where *MIR164C* plays a specific role in regulation of petal number [68].

How *MIR172* family members are differentially regulated by SPLs in Arabidopsis is poorly understood. First, although the *SPL* mRNA level is extremely low, miR172 is still detectable in the juvenile phase. This result suggests that some unknown transcription factors contribute to the basal level of miR172 at this developmental stage. It is unlikely that this putative transcription factor activates miR172 expression through *MIR172E* because our genetic and reporter analyses showed that *MIR172E* is largely not functional under normal growth conditions. Second, it is also unclear why *MIR172A*, *MIR172B*, and *MIR172C* are highly expressed in the leaf vascular tissues. It is possible that this tissue specificity is generated by transcriptional activators, including SPLs (SPL9, SPL10, and SPL11) and GI, which are predominantly expressed in the vascular tissues [63,67]. Third, we have shown that the SPL15-*MIR172D* pair regulates floral transition in the SAM (Fig 8). Interestingly, the predominant expression of *SPL15* in the SAM is conserved within species in the Brassicaceae [69]. Therefore, identification and analysis of the *cis*-elements that confer this tissue specificity is an important future research

direction. It also remains unknown why SPL9 or SPL10 are not able to activate *MIR172D* in leaves. One possibility is that the *MIR172D* locus is transcriptionally competent only in the SAM. In this scenario, SPL15 would activate *MIR172D* with the help of a cofactor which is exclusively expressed in the meristem. A promising candidate is FD, which is expressed only in the SAM and induces flowering through its interaction with FT [70,71]. Moreover, in support of this hypothesis, the ATAC-seq datasets have revealed that the *MIR172D* locus is largely inaccessible in the 1st and 7th leaves (S6 Fig).

Previous reports have shown that the miR172 level varies in response to the ambient temperature [65,66,72]. Interestingly, the data presented in this manuscript show that, in addition to being regulated posttranscriptionally by FCA, environmental temperature can influence miR172 levels by modulating at least 2 *MIR172* genes (*MIR172C* and *MIR172D*) at the transcriptional level [66]. The molecular link between temperature sensing and *MIR172* transcription is unknown at present. Genetic analyses have demonstrated that 2 MADS-box floral repressors SHORT VEGETATIVE PHASE (SVP) and FLOWERING LOCUS M (FLM) play critical roles in flowering time at 16˚C [73–76]. While FLM is regulated by temperature-dependent alternative splicing [77–79], SVP is subjected to temperature-dependent destabilization [80]. The FLM-SVP dimer represses flowering through *FT* and *SOC1* [80]. Therefore, it is plausible to assume that the FLM-SVP complex may bind to the promoter of *MIR172C* and *MIR172D* and suppress their expression at 16˚C. Genome-wide mapping of the binding sites of the FLM-SVP complex and identification of the temperature responsive *cis*-elements in the *MIR172C* and *MIR172D* promoters will be necessary to test this hypothesis in the future.

Our findings have useful implications with respect to the regulation of flowering time in crops. Genome-wide analyses have found that, as in Arabidopsis, the *MIR172* genes are present as multigene families in the rice, maize, and wheat genomes [81–86]. The functions of the individual *MIR172* genes in flowering and floral patterning is largely unexplored. The Hake lab has shown that *tasselseed4* (*ts4*) is encoded by a *MIR172* gene in maize [87]. The mutation in *ts4* permits carpel development in the tassel while increasing meristem branching. Thus, this result implies that *MIR172* genes may also undergo functional specification in some crop species. Undoubtedly, careful comparisons between wild species and cultivars will shed light on whether the natural variations in SPL-*MIR172* pairs underlie the evolution of plasticity in flowering time that has occurred during crop domestication.

In conclusion, our results reveal that the expansion of the *MIR172* gene family in the Arabidopsis genome provides molecular substrates for the integration of diverse floral inductive cues including age, photoperiod, and temperature. The matching pairs of coexpressed SPL and *MIR172* genes ensure the precise acquisition of floral competence under different conditions in order to maximize seed yields.

## Methods

### Plant materials and growth conditions

*A. thaliana* (ecotype Col-0) plants were grown at 22˚C (day)/19˚C (night) in long days (16 hours light/8 hours dark) or at 21˚C (day)/18˚C (night) short days (8 hours light/16 hours dark). For growth under natural conditions, plants were sowed on the soil and grown in the pots (5 × 5 cm) in Shanghai (SIPPE, Fenglin campus) from December 2018 to May 2019. The average temperature was 10˚C (day)/5˚C (night). The day length varied from 12 hours light/12 hours dark (December 2018) to 14 hours light/10 hours dark (May 2019). For temperature assays, the plants were grown in the chamber (Percival) at 16˚C or 28˚C in long days. For transgenic *A. thaliana* plants, the binary constructs were delivered into *Agrobacterium tumefaciens* strain GV3101 (pMP90) by freeze-thaw method. Transgenic plants were generated by

floral dipping method [88] and screened with 0.05% glufosinate (Basta) on soil, 40 μg/ml hygromycin or 50 μg/mL kanamycin on half-strength Murashige and Skoog (MS) media.

## Constructs

The oligonucleotide primers for all the constructs are given in S3 Table. The SnapGene map for each construct is available upon request. To generate *MIR172* reporters, the promoter regions of *MIR172* genes were PCR amplified and cloned in front of *GFP-N7* or *3xVenus-N7* coding region in the binary vector VV00 (S6 Fig).

## Generation of *mir172* mutants by CRISPR-Cas9

Gene-targeting vectors were constructed based on the pHEE2A-TRI (rbcS-E9t) system provided by Dr. Qi-Jun Chen [49]. The sgRNA sequences for each *MIR172* gene were designed using tools at CRISPR-P 2.0 (http://crispr.hzau.edu.cn/CRISPR2/) according to website instructions [89]. The oligonucleotide primers for sgRNAs are given in S3 Table. The mutants were identified by PCR. The mutants were backcrossed to WT to remove transgene and potential off-target mutations. The high-order mutants were generated by crossing and PCR-based genotyping.

## Phenotypic analyses

For flowering time measurement, the seeds of different genotypes were sowed on soil. The plants were grown under long-day or short-day conditions. Because the leaf initiation rate of the *mir172* mutants was comparable to that of WT, the flowering time was directly measured by counting the total number of rosette leaves when plants started bolting (10 cm in height). The branches on the primary bolt and the average internode length were scored after the inflorescence SAMs were fully terminated. The floral patterning phenotype were examined using Olympus BX63 equipped with DP73 digital camera.

## Statistical analyses

We used ordinary one-way ANOVA to perform statistical analyses. Brown–Forsythe test and Bartlett test were performed whenever multiple samples were compared. Statistical significance was determined at $p < 0.05$ unless otherwise indicated. Statistical test results are given in S4 Table. The original data were given in S1 and S2 Data and deposited in https://figshare.com/articles/dataset/202011_Lian_PLOS_B_Original_data_xlsx/13293722.

## Microscopy

The inflorescence SAM samples were processed and scanned using JSM-6360LV Scanning Electron Microscope (SEM, JEOL). The SEM images were analyzed with Image J software as described [90,91].

Phenotypic analyses and imaging of floral organs and meristems were examined using Olympus BX63 equipped with DP73 digital camera as described [90,91]. For RNA in situ imaging, slides were mounted with water as described and observed under Olympus BX63 equipped with DP73 digital camera and differential interference contrast module. To compare the *MIR172* reporter activities between WT and the *spl* mutants, the *MIR172* reporter line was crossed to the *spl9* or *spl15* mutant. The homozygous mutants were identified in F2 and used for analyses.

For confocal imaging, the flowers and inflorescences were dissected on 2% agar plate under a stereo microscope. For the vegetative samples collected from seedlings, the tissues were

collected and immediately placed in the vials with ice-cold phosphate buffered saline (PBS) containing 2.5% paraformaldehyde (PFA (pH 7.0)). The samples were infiltrated for 30 minutes by vacuum and stored at 4°C overnight. Tissues were then washed with sucrose gradient PBS-PFA solution, embedded with 6% low melting agarose, and sliced with a Lecia Sliding Microtome 1200S at the thickness of 50 μm. For the vegetative samples collected from adult plants, the shoot apices were dissected by free-hand sectioning under a stereo microscope. Selected sections were stained in 10 μM FM4-64 solution (Sigma, Merck, China) or mounted directly with water without staining.

Prepared specimen was observed and scanned with Olympus FV3000 or Leica SP8 confocal microscope. Proper filter sets and lasers were selected for fluorescence signal scanning. All specimens were scanned with 10× objective (HCX PL APO CS 10X/0.40 DRY on Leica SP8 or UPLXAPO 10X/0.40 on Olympus FV3000). For GFP, excitation light wave length was 488 nm; emission, 500 to 550 nm. For Venus, excitation light wave length was 415 nm (on Leica SP8) or 488 nm (on Olympus FV3000); emission, 520 to 550 nm. For chlorophyll and FM4-64, excitation light wave length was 415 nm (on Leica SP8) or 488 nm (on Leica SP8 or Olympus FV3000), emission, 650 to 750 nm. The same settings for visualizing GFP or Venus were used for each sample. The settings for visualizing plant cell walls by FM4-64 was modified for presentation purposes. The interpretation of the results was not affected, and the original images are available upon request.

Quantification of reporter intensity (S8 Fig; S10B Fig) was performed using Image J software. Briefly, the intensities of GFP or Venus in the selected area (vascular tissues or the SAM) were measured. For each *MIR172* reporter line, the images taken from 3 to 6 individual plants were measured.

## Expression analyses

Total RNA was extracted from seedlings, leaves, roots (harvested from 7-day-old seedlings), or shoot apices with Trizol reagent (ThermoFisher, Cat No./ID: 15596018). A total of 1 μg of total RNA was DNase I-treated (1 unit/mL; ThermoFisher, Cat No./ID: EN0521) and used for cDNA synthesis with oligo (dT) primer and/or miR172-RT-Primer A primer using RevertAid RT Reverse Transcription Kit (ThermoFisher, Cat No./ID: K1691) (S14A Fig). The average expression levels and standard errors were calculated from $2^{-\Delta\Delta Ct}$ values. Two or three biological replicates were performed. For each biological replicate, 2 technical replicates were performed. The qRT-PCR primers for *TUB* have been described [30]. The oligonucleotide primers for all the genes are given in S3 Table. qRT-PCR on mature miR172 was performed as described [92,93]. The difference among miR172 isoforms resides in the last nucleotide (S14A Fig). The miR172-RT-Primer A primer used in this study did not distinguish these isoforms (S14B Fig).

## RNA in situ hybridization

Shoot apices from short-day grown plants of different ages were dissected and fixed in formalin:acetic acid:ethanol (1:1:18). Paraffin-embedded materials were sectioned to 8 μm thickness. RNA in situ hybridization was performed as described [62,90]. For *SPL2* and *SPL10* probes, the cDNA fragments of *SPL2* and *SPL10* were PCR amplified and cloned into pBluescript SK, respectively. In vitro transcription was performed with T3 or T7 RNA polymerase (Thermo-Fisher, Cat No./ID: EP0101/EP0111), in which linearized vectors were used as templates. For miR172 probe, Locked Nucleic Acid (LNA) oligonucleotide was end labeled with the DIG oligonucleotide 3′-end labeling kit (Roche, Cat No./ID: 03 353 575 910).

## Accession numbers

Sequence data from this article can be found in the Arabidopsis Genome Initiative or Gen-Bank/EMBL databases under the following accession numbers: *MIR172A* (At2g28056), *MIR172B* (At5g04275), *MIR172C* (At3g11435), *MIR172D* (At3g55512), *MIR172E* (At5g59505), *SPL2* (At5g43270), *SPL9* (At2g42200), *SPL10* (At1g27370), and *SPL15* (At3g57920).

## Supporting information

**S1 Fig. Characterization of the *mir172* CRISPR-Cas9 mutants.** The genomic sequence of WT and mutated *MIR172* loci. Orange, stem-loop of *MIR172*; green, miRNA*; blue, miRNA. CRISPR, clustered regularly interspaced short palindromic repeats; miRNA, microRNA; WT, wild type.
(TIF)

**S2 Fig. Expression of miR172 and its targets in the *mir172* mutants.** (A) Expression of miR172 in different tissues in WT. Seven-day-old seedlings, the 6th and 8th leaves, roots, and inflorescence were used. Two technical replicates for each biological replicate (*n* = 2) were performed. Error bars represent SD. (B) Expression of miR172 in the *mir172* mutants. Twelve-day-old plants in long days were used for qRT-PCR analyses. Two technical replicates for each biological replicate (*n* = 2) were performed. Error bars represent SD. (C to G) Expression of miR172 targets in the *mir172* mutants. Twelve-day-old plants in long days were used for qRT-PCR analyses. Two technical replicates were performed. Error bars represent SD. (H to L) Time course analysis of abundance of miR172 targets. Seven-day, 14-day, and 21/25-day-old plants in long days were used for qRT-PCR analyses. Two technical replicates for each biological replicate (*n* = 2) were performed. Error bars represent SD. For all the qRT-PCR assays, the expression level was normalized to that of *TUB*. The data underlying this figure are included in S2 Data. qRT-PCR, quantitative real-time PCR; *TUB*, *β-TUBULIN-2*; WT, wild type.
(TIF)

**S3 Fig. Phenotype of the *mir172abcde* mutants grown under outdoor conditions.** The photos were taken after plants flowered. Scale bar represents 1 cm.
(TIF)

**S4 Fig. Abaxial trichome phenotype.** WT and the *mir172* mutants were grown at 22˚C in long days. The abaxial surfaces of the leaves are shown. Scale bar represents 200 μm.
(TIF)

**S5 Fig. Floral phenotype of the *mir172* mutants.** (A) The flowers of WT and the *mir172* mutants. The flowers with abnormal phenotypes are shown. Scale bars represent 500 μm. (B) Quantification of abnormal floral phenotype in different genotypes. The floral phenotype of 5 to 21 plants for each genotype was examined. The data underlying this figure are included in S2 Data.
(TIF)

**S6 Fig. Generation of *MIR172* reporters.** The ATAC-seq tracks for the *MIR172* genes are shown. The datasets are derived from 4 plant tissues including the 1st leaf, the 7th leaf, embryo, and seedlings [56]. The orange box and line indicate stem loop region and the sequences used for generation of *MIR172* reporter, respectively. For *MIR172E* reporter, the 1.3 kilobase pair (kb) downstream sequence was also included. The ATAC-seq datasets are deposited in Beijing Institute of Genomics Data Center (http://bigd.big.ac.cn) with the accession number

(BioProject PRJCA002620 and BioProject PRJCA003872). The data underlying this figure are included in S2 Data.
(TIF)

**S7 Fig. Expression of *MIR172E* reporter in long days.** The leaves (A) and shoot apices (B) are shown. Plants were grown at 22˚C in long days. The same pinhole size was used for scanning. Dash line marks the SAM. Scale bars represent 100 μm.
(TIF)

**S8 Fig. Quantification of *MIR172* reporter genes in long days.** Plants were grown at 22˚C in long days. See also Fig 4. The data underlying this figure are included in S2 Data.
(TIF)

**S9 Fig. Expression of *MIR172* reporter genes in the *spl9* and *spl15* mutants.** (A to C) Expression of *MIR172* reporter genes in the *spl9* and *spl15* mutants. Plants were grown at 22˚C in long days. Please note that the promoter activity of *MIR172C* was attenuated in the *spl9* mutant but not in the *spl15* mutant (B). In contrast, the transcription of *MIR172A* was largely unaffected by the mutation in *SPL9* or *SPL15* (A). The promoter activity of *MIR172D* was decreased in the *spl15* mutant (C). The same confocal settings were used for scanning for each reporter line. Dash line marks the SAM. Scale bars represent 100 μm. (D) Expression of *MIR172* genes in 18-day-old *spl9* and *spl15* mutants grown in LD. We harvested plants with the cotyledons and the first 5 rosette leaves manually removed. We could not get faithful data for *MIR172C* because its transcript level was very low. The expression level in WT is set to 1.0. Expression was normalized to *TUB*. Two technical replicates for each biological replicate (*n* = 2) were performed. Error bars represent SD. The data underlying this figure are included in S2 Data. LD, long days; SAM, shoot apical meristem; *TUB*, *β-TUBULIN-2*; WT, wild type.
(TIF)

**S10 Fig. Expression of *MIR172* genes and *MIR172* reporter genes in short days.** (A) qRT-PCR analyses of *MIR172* genes. Due to low expression levels of *MIR172C* and *MIR172D* in short days, their expression were not faithfully quantified. Two technical replicates for each biological replicate (*n* = 2) were performed. Error bars represent SD. See also Fig 7. (B) Quantification of *MIR172* reporter genes. See Fig 6. The data underlying this figure are included in S2 Data.
(TIF)

**S11 Fig. Expression of *SPL2* and *SPL10* in shoot apices.** Scale bars represent 50 μm.
(TIF)

**S12 Fig. Expression of *MIR172* reporter genes at different temperatures.** Expression of *MIR172A* (A) and *MIR172B* (B) in leaf and shoot apices. Plants were grown at 16˚C or 22˚C in long days. The same confocal settings were used for scanning for each reporter line. Dash line marks the SAM. Scale bars represent 100 μm.
(TIF)

**S13 Fig. Expression of *MIR172* reporter genes in response to photoperiod.** (A to D) The expression of *MIR172* reporter genes before and after shift. Plants were grown in short days for 3 weeks (SD 3w) and shifted to long days (shift + n days). (E) The expression of *MIR172* reporter genes after 4 weeks in short days (SD 4w). Please note that all the plants were still in the vegetative phase and the expression pattern of each reporter was the same as that of 3-week-old plants. The same confocal settings were used for scanning for each reporter line.

Dash line marks the SAM. Scale bars represent 100 μm.
(TIF)

**S14 Fig. Validation of qRT-PCR primers.** (A) The comparison of miR172 isoforms. The difference among miR172 isoforms resides in the last nucleotide (red). Two miR172-RT primers are designed: The miR172-RT-Primer A is fully complementary to miR172a, miR172b, and miR172e, whereas the miR172-RT-Prime C is fully complementary to miR172c and miR172d. These RT primers bind to the 3′ portion of miR172 molecules (underlined), initiating reverse transcription. (B) The expression of miR172 in WT and the *mir172* mutants. We harvested 14-day-old plants grown in long days and set up 2 qRT-PCR experiments using miR172-RT-Primer A (left) or miR172-RT-Primer C (right) as the RT primer, respectively. Since *MIR172D* is highly expressed in the SAM, we could not detect its contribution to the mature miR172 pool. As such, the abundance of miR172 is markedly decreased in the *mir172abe* mutants. Both RT primers gave rise to similar results. However, the miR172-RT-Primer A gave higher reverse transcription efficiency than miR172-RT-Primer C. Notably, we could still detect miR172 in the *mir172cd* mutant using miR172-RT-Primer A as the RT primer, indicating that this primer does not discriminate miR172 isoforms. Expression was normalized to *TUB*. Two technical replicates for each biological replicate (*n* = 2) were performed. Error bars represent SD. The data underlying this figure are included in S2 Data. qRT-PCR, quantitative real-time PCR; SAM, shoot apical meristem; *TUB*, *β-TUBULIN-2*; WT, wild type.
(TIF)

**S1 Table. Flowering time results.**
(XLSX)

**S2 Table. List of the *mir172* mutants used in this study.**
(XLSX)

**S3 Table. List of the primers and probe used in this study.**
(XLSX)

**S4 Table. Statistical test results.**
(XLSX)

**S1 Data. The data underlying main figures.** From Figs 1D, 2B, 2D, 2F, 2H, 3B–3D, 4E–4J, 5B, 5D, 7A–7E.
(XLSX)

**S2 Data. The data underlying supporting figures.** From Figs S2A–S2G, 5, 6, 8, 9D, 10A, 10B, 14B.
(XLSX)

## Acknowledgments

We thank Ji-Qin Li, Zhi-Ping Zhang, and Xiao-Yan Gao (SIPPE, CAS) for skillful technical assistance; George Coupland (Max Planck Institute for Plant Breeding Research, Germany) and members in J.-W. Wang lab for discussion and comments on the manuscript.

## Author Contributions

**Investigation:** Heng Lian, Long Wang, Ning Ma, Chuan-Miao Zhou, Lin Han, Tian-Qi Zhang.

**Supervision:** Jia-Wei Wang.

**Writing – original draft:** Jia-Wei Wang.

**Writing – review & editing:** Jia-Wei Wang.

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
