## [Editor Report · Decision Letter 0]

14 Jul 2020

Dear Dr Wang, 

Thank you for submitting your manuscript entitled "Redundant and Specific Roles of Individual MIR172 Coding Genes in Plant Development" for consideration as a Research Article by PLOS Biology.

Your manuscript has now been evaluated by the PLOS Biology editorial staff as well as by an academic editor with relevant expertise and I am writing to let you know that we would like to send your submission out for external peer review.

Please re-submit your manuscript within two working days, i.e. by Jul 16 2020 11:59PM.

Kind regards,

Ines

--

Ines Alvarez-Garcia, PhD

Senior Editor

PLOS Biology

---

## [Decision Letter · Decision Letter 1]

23 Sep 2020

Dear Dr Wang,

Thank you very much for submitting your manuscript "Redundant and Specific Roles of Individual MIR172 Coding Genes in Plant Development" for consideration as a Research Article at PLOS Biology. Thank you also for your patience as we completed our editorial process, and please accept my sincere apologies for the delay in providing you with our decision. Your manuscript has been evaluated by the PLOS Biology editors, an Academic Editor with relevant expertise, and by three independent reviewers.

The reviews are attached below. You will see that the reviewers find your results interesting and novel and think it is worth pursuing publication of the manuscript in PLOS Biology. Thus we are pleased to offer you the opportunity to address the points raised by the reviewers in a revised version that we anticipate should not take you very long. We will then assess your revised manuscript and your response to the reviewers' comments and we may consult the reviewers again.

We expect to receive your revised manuscript within 1 month.

**IMPORTANT - SUBMITTING YOUR REVISION**

*Resubmission Checklist*

*Published Peer Review*

*PLOS Data Policy*

*Blot and Gel Data Policy*

Sincerely,

Ines

--

Ines Alvarez-Garcia, PhD,

Senior Editor,

ialvarez-garcia@plos.org,

PLOS Biology

Reviewers’ comments

Rev. 1:

This manuscript presents an analysis of redundant and/or specific functions of MIR172 gene family regulating plant development such as flowering time, axillary branching, meristem size and trichome development. The authors generate single mutants of each MIR172 A/B/C/D/E gene by CRISPR-Cas9 technology and analyze developmental phenotypes of a single mutant and higher-order mutants to address the redundant and/or specific functions of the each MIR172 gene. The authors further dissect the function of MIR172 genes in flowering time through genetic analysis using mir172 mutant plants and MIR172 gene promoter-driven reporter lines, showing that spatiotemporal expression of MIR172 gene family contributes to floral transition. Furthermore, in response to photoperiod, diversified function of MIR172 gene family is shown to be coupled with specific SPL gene possibly in a spatiotemporal specific manner.

Most of plant miRNAs are encoded by gene families but specific or redundant function of miRNA members has not been systemically addressed. miRNAs are thought to locally or distally act as signaling molecules and this study clearly shows that spatiotemporal activities of individual miR172 displays differential responses to developmental and environmental cues. Moreover, distinct pairs of SPL and miR172 regulate flowering time in response to photoperiod, suggesting that evolutionarily diversified miRNA family contribute to the internal and external signaling complexity in plants. This is a very well-organized and well-written paper, which can be a good example of how to study the diversified biological functions of plant MIRNA gene family. I have only a few minor comments.

1. How were the reporter lines in spl9 and spl15 (Fig. 4J and Fig. 6F) generated? Were they crossed to those mutants?

2. Following the first comments, it would be nice to determine expression levels of MIR172A-E in spl9 (fig. 4J) and spl15 (Fig. 6F) mutants to confirm the reporter line-based observation.

Rev. 2: Tony Millar – this reviewer has waived anonymity

This is a very comprehensive study. Very impressively, they have used CRISPR to generate a quintuplet mutant in miR172, generating a very strong loss-of-function miR172 mutant that displays phenotypes they have not been previously observed from other approaches, such as using MIM172 transgenic plants. This has enable them to gain a greater insight into the role of miR172 in plant development, and has generated a very powerful genetic resource of mutant combinations to study the role of individual miR172 gene in the model plant Arabidopsis. The vast amount of data generated in this study has been very well presented into logical well thought-out figures.The paper gives insight into the contributions of these individual miR172 genes in different developmental processes, mainly that of the different cues of flowering-time.

My major comment is more technical that that relating to the biology uncovered by the study. The structure of the reporter genes the authors made it not entirely clear to me. From Sup Fig 6, which covered the regions that were used in the construct, it seems that the stem-loop regions of the miR172 genes were also included, not just the 5' upstream regions (authors call this promoters in the Materials and methods). If these stem-loop regions are then in the same transcript as the reporter (GFP or Venus) would not then the transcript be processed by DCL1 and interfere with the reporter gene expression and potentially alter the pattern? I think this is an important consideration given the extent to which these reporters are used in the study.

Minor comments

Continuing on regarding the reporter lines, only one line per construct was examined in detail "We chose one representative T2 line for the subsequent analyses." - which "gave consistent and reliable expression patterns". Authors need to be clearer here, how do they know that the line picked was consistent and reliable? Does this mean of the 20 lines screened, all have highly similar expression patterns, so only one was chosen for detailed analysis? If so please state. As variation between lines is possible (as they are independent transgenic events), so was the one line chosen representative of the general expression pattern observed most of the 20 lines? i.e. for MIR172d -did a majority of the 20 share the observation that they are expressed in leaf margins. If this is the case, I think the authors need to add a few words for a clearer explanation. If not this would also be of concern.

Conflicting statements that need to be clarified "the five MIR172 gene family members are functionally redundant but exert individual specificities in regulating diverse aspects of plant development" - but in the discussion - "we have demonstrated that the Arabidopsis MIR172 gene family members are definitely not redundant" - think the former statement is more accurate and the latter misleading - they may not be fully redundant, but they play overlapping roles in some instances. Please clarify.

Figure 3C. Here it appears mature miR172 levels are normalized to TUB, a mRNA not a small RNA?? Also, miR172a and b, are the same isoform, whereas the others have sequence variations. Are the primers used preferentially detect the a/b isoform and therefore skew the data. Need more information here, the isoforms, the primer used (any mis-matches with c, d and e?), and why TUB was used (polyA synthesis?) versus small RNA Reverse Transcription with a specific primer??

Phrases/ typos

Page 3, line 11; MIR319 should be MIR159 as indicated in the next sentence.

Page 5 line 19 and other places; Using the word "coding" regarding a miRNA gene could be regarded as incorrect given they are usually classified as "non-coding RNA". E.g. "the entire stem-loop regions of the "coding" genes" - maybe better to say "MIR172 genes"

Rev. 3:

Revision of "Redundant and Specific Roles of Individual MIR172 Coding Genes in Plant Development" by Heng Lian et al.

This work analyzed the MIR172 family of small RNAs in Arabidopsis thaliana, which is comprised by five members. The authors used CRISPR-Cas9-induced mutations to inactivate each MIR172 family member. The authors characterize diverse phenotypic defects after the inactivation of different combinations of MIR172 genes, including the modification of meristem size, trichome initiation, stem elongation, shoot branching and floral competence. Furthermore, the authors demonstrate the role of specific SPL-MIR172 pairs in different tissues.

The generation of mutants for all MIR172 family members and the thorough and laborious genetic analysis performed here are very well done and I think that the results will be a of wide interest to the scientific community. Overall, the results are presented in a clear way highlighting the role of MIR172s as a hub for the integration of different signals.

My main comment about the manuscript is that it is lacking a concomitant description about the expression miR172-targets in wt and mir172 mutants. At least, the expression of miR172-targets could be determined by RT-qPCR in the samples described in Figure 3C or S2B.

Additional comments:

1) The reporters for MIRNA expression were designed by replacing the precursor sequence with a reporter. I wonder whether this strategy could result in a reporter mRNA having long 5' UTR with spurious ATGs that will in turn affect its expression. Please comment.

2) Title "MIR172 coding genes". I think it should be "miR172-coding genes". Anyways, I'm not sure whether it is convenient to use "coding".

3) Introduction: "Due to the small sizes of the genes, simultaneous inactivation of all miRNA family members by generation of multiple transfer-DNA (T-DNA) mutant lines has so far been achieved for only two relatively small families, MIR164 (i.e. MIR164A-C) and MIR319 (i.e. MIR319A-C) [11, 12]" -- It is MIR59A-B instead of MIR319A-C.

4) Introduction: "These two studies showed that, similar to protein-coding genes, the MIRNA coding genes in the same family are functionally redundant." -- MIR164C (early extra petals1) has also specific functions (Baker et al. Curr Biol 2005)

5) I think Figure S1 is nicer than Figure 1. Perhaps the authors might want to move Figure S1C to Figure 1.

6) Page 13 l6 "promoting the procession" - I think there is a typo "promoting the processing"

---

## [Editor Report · Decision Letter 2]

23 Nov 2020

Dear Dr Wang,

Thank you for submitting your revised Research Article entitled "Redundant and Specific Roles of individual MIR172 Genes in Plant Development" for publication in PLOS Biology. I have obtained advice from the Academic Editor and have discussed the revision with the editorial team. 

We're delighted to let you know that we're now editorially satisfied with your manuscript. However before we can formally accept your paper and consider it "in press", we also need to ensure that your article conforms to our guidelines. A member of our team will be in touch shortly with a set of requests. As we can't proceed until these requirements are met, your swift response will help prevent delays to publication. Please also make sure to address the data and other policy-related requests noted at the end of this email.

- a cover letter that should detail your responses to any editorial requests, if applicable

*Copyediting*

*Published Peer Review History*

*Early Version*

Sincerely,

Ines

--

Ines Alvarez-Garcia, PhD

Senior Editor,

PLOS Biology

Fig. 1D; Fig. 2B, D, F, H; Fig. 3B-D; Fig. 4E-J; Fig. 5B, D; Fig. 7A-E; Fig. S2A-L; Fig. S5B; Fig. S6A-E; Fig. S8A-D; Fig. S9D; Fig. S10A, B and Fig. S14B

---

## [Editor Report · Decision Letter 3]

10 Dec 2020

Dear Dr. Wang,

I am writing concerning your manuscript submitted to PLOS Biology, entitled “Redundant and Specific Roles of Individual MIR172 Genes in Plant Development.”

We have now completed our final technical checks and have approved your submission for publication. You will shortly receive a letter of formal acceptance from the editor.

Kind regards,

PLOS Biology